# Recommendations for the Management of Patients with Hairy-Cell Leukemia and Hairy-Cell Leukemia-like Disorders: A Work by French-Speaking Experts and French Innovative Leukemia Organization (FILO) Group

**DOI:** 10.3390/cancers16122185

**Published:** 2024-06-10

**Authors:** Jérôme Paillassa, Elsa Maitre, Nadia Belarbi Boudjerra, Abdallah Madani, Raihane Benlakhal, Thomas Matthes, Eric Van Den Neste, Laura Cailly, Luca Inchiappa, Mohammed Amine Bekadja, Cécile Tomowiak, Xavier Troussard

**Affiliations:** 1Service des Maladies du Sang, CHU d’Angers, 49000 Angers, France; jerome.paillassa@chu-angers.fr; 2Hématologie Biologique, Structure Fédérative D’oncogénétique Cyto-Moléculaire du CHU de Caen (SF-MOCAE), CHU de Caen, 14000 Caen, France; maitre-e@chu-caen.fr; 3Unité MICAH, INSERM1245, Université Caen-Normandie, 14000 Caen, France; 4Service d’Hématologie, CHU Béni Messous, Alger 16308, Algeria; nboudjerra@hotmail.fr; 5Service d’Hématologie, CHU de Casablanca, Casablanca 20000, Morocco; madani.hemato@gmail.com; 6Service d’Hématologie, CHU Aziza Othmena, Tunis 1002, Tunisia; raihane.benlakhal@gmail.com; 7Service d’Hématologie, Département d’Oncologie et Service de Pathologie Clinique, Département de Diagnostic, Hôpital Universitaire de Genève, 1205 Genève, Switzerland; thomas.matthes@hcuge.ch; 8Cliniques Universitaires Saint-Luc, Université Catholique de Louvain, 1000 Brussels, Belgium; eric.vandenneste@saintluc.uclouvain.be; 9Service d’Onco-Hématologie et de Thérapie Cellulaire, CHU de Poitiers, 86000 Poitiers, France; laura.cailly20@gmail.com (L.C.);; 10Service d’Hématologie, Institut Paoli-Calmette, 13397 Marseille, France; 11Service d’Hématologie et de Thérapie Cellulaire, EHU Oran, Oran 31000, Algeria; mabekadja@yahoo.fr; 12Hematologie CHU Caen Normandie, 14000 Caen, France

**Keywords:** hairy-cell leukemia, HCL, hairy-cell leukemia variant, HCL-V, splenic diffuse red pulp lymphoma, SDRPL, diagnosis, treatment, recommendations, flow cytometry, BRAF*^V600E^* mutation, BRAF inhibitors

## Abstract

**Simple Summary:**

The diagnosis of hairy-cell leukemia (HCL) and HCL-like disorders including the variant form of HCL (HCL-V), splenic diffuse red pulp lymphoma (SDRPL) and splenic marginal zone lymphoma (SMZL) is a challenge in clinical practice. We discuss the major points for the diagnosis of HCL and HCL-like disorders and we propose recommendations for the diagnosis of HCL, treatment in first line and in relapsed/refractory patients.

**Abstract:**

Introduction: Hairy-cell leukemia (HCL) is a rare B-cell chronic lymphoproliferative disorder (B-CLPD), whose favorable prognosis has changed with the use of purine nucleoside analogs (PNAs), such as cladribine (CDA) or pentostatin (P). However, some patients eventually relapse and over time HCL becomes resistant to chemotherapy. Many discoveries have been made in the pathophysiology of HCL during the last decade, especially in genomics, with the identification of the BRAF*^V600E^* mutation and cellular biology, including the importance of signaling pathways as well as tumor microenvironment. All of these new developments led to targeted treatments, especially BRAF inhibitors (BRAFis), MEK inhibitors (MEKis), Bruton’s tyrosine kinase (BTK) inhibitors (BTKis) and recombinant anti-CD22 immunoconjugates. Results: The following major changes or additions were introduced in these updated guidelines: the clinical relevance of the changes in the classification of splenic B-cell lymphomas and leukemias; the increasingly important diagnostic role of BRAF*^V600E^* mutation; and the prognostic role of the immunoglobulin (IG) variable (V) heavy chain (H) (*IGHV*) mutational status and repertory. We also wish to insist on the specific involvement of bones, skin, brain and/or cerebrospinal fluid (CSF) of the disease at diagnosis or during the follow-up, the novel targeted drugs (BRAFi and MEKi) used for HCL treatment, and the increasing role of minimal residual disease (MRD) assessment. Conclusion: Here we present recommendations for the diagnosis of HCL, treatment in first line and in relapsed/refractory patients as well as for HCL-like disorders including HCL variant (HCL-V)/splenic B-cell lymphomas/leukemias with prominent nucleoli (SBLPN) and splenic diffuse red pulp lymphoma (SDRPL).

## 1. Introduction

HCL is considered as part of splenic B-cell lymphoma/leukemia in the 5th edition of the World Health Organization (WHO) classification of haematolymphoid tumours (WHO-HAEM5) along with splenic marginal zone lymphoma (SMZL), splenic diffuse red pulp lymphoma (SDRPL) and splenic B-cell lymphoma with prominent nucleoli (SBLPN). SBLPN includes the variant form of HCL (HCL-V), CD5 negative B-cell prolymphocytic leukemia (B-PLL) and some SMZL with prominent nucleoli. Distinguishing all these entities is challenging in daily clinical practice because of the different clinical courses and the need for different treatments. The integration of clinical, morphological, histological, phenotypical and molecular data can help to establish the final diagnosis, particularly in complex cases.

Originally described in 1958 by Bouroncle et al., hairy-cell leukemia (HCL) is an indolent and rare B-cell chronic lymphoproliferative disorder (B-CLPD) [1]. The morphological identification of the hairy cells (HCs) in the peripheral blood (PB), bone marrow (BM) and/or spleen (S) must be completed by their characterization by multiparametric flow cytometry (FCM). The trephine BM biopsy is necessary to quantify the tumor burden and complete the diagnosis using annexin A1 expression [2]. Important discoveries have been made during the last decade in genomics [3,4] and cellular biology including the signaling pathways [5] and the tumor microenvironment [6]. The BRAF*^V600E^* mutation is considered as a disease-defining event [7] and has been identified in more than 90% of HCL patients. Before the 1990s, HCL patients were treated with splenectomy [8] or interferon alpha-2a (IFNα-2a) [9]. They achieved partial and short-lasting responses and often died from progression and/or infections. Purine nucleoside analogs (PNAs) such as cladribine (CDA) or pentostatin (P) were introduced later and are routinely used for HCL treatment. The prognosis is favorable with prolonged responses and survival [10,11,12,13]. In case of relapses, the repeated use of PNAs can be dangerous and leads to sometimes fatal infectious complications and/or secondary malignancies. The HCL gradually becomes refractory to PNAs after several courses of treatment [12,14,15,16].

New treatments have recently emerged in HCL: BRAF inhibitors (BRAFis) (vemurafenib, dabrafenib) [17,18,19] associated or not with MEK inhibitors (MEKis) (trametinib, binimetinib) [20] or anti-CD20 monoclonal antibodies (rituximab, obinutuzumab), Bruton’s tyrosine kinase inhibitors (BTKis) (ibrutinib, zanubrutinib) [21,22] and recombinant anti-CD22 immunoconjugates (Moxetumomab pasudotox (Moxe)) [23].

We previously published recommendations for the first time in 2014, before all these discoveries and the results of important clinical trials [24]. To update these recommendations, we organized several meetings with French-speaking experts (Algeria, Belgium, France, Morocco, Switzerland, Tunisia) in HCL and members of the French Innovative Leukemia Organization (FILO) group in order to update these recommendations, taking into account the new targeted therapies and integrating them in a new therapeutic algorithm. We also propose recommendations for the diagnosis of HCL and HCL-like disorders including HCL-V, SDRPL and SMZL.

## 2. Hairy-Cell Leukemia



**Epidemiology of HCL**




-HCL accounts for 2–3% of all leukemias [25,26,27] and fewer than 1% of all hematological disorders.-The incidence rate was 0.3 per 100,000 for HCL and 0.2 per 100,000 for HCL-V in the United States [28]. It was lower in non-Hispanic/Blacks, Hispanics, and Asians/Pacific Islanders. In 2018, the number of new cases was estimated at 300 in France. Incidence between 1990 and 2018 remained relatively stable despite the aging of the population and better access to diagnostic tools [29].-The male-to-female sex ratio is 5:1.-At diagnosis, the median age is 63 years for men and 59 years for women.-The etiology of HCL remains largely unknown but several risk factors have been suspected: farming, exposure to pesticides, petroleum derivatives, diesel and ionizing radiation [30,31,32,33]. Tobacco appears to be protective for HCL [34]. Rare familial cases have also been described.




**HCL Diagnosis**



The symptoms of HCL are related to BM failure (severe or recurrent infections, anemia, thrombocytopenia with hemorrhages), spleen involvement or fortuitous identification of HC during a routine complete blood count (CBC). Splenomegaly is common, while hepatomegaly and enlarged lymph nodes are rare. In about 20% of cases, opportunistic infections occur at diagnosis [12] and include mycobacterioses, *Listeria monocytogenes* infection, *Toxoplasma gondii* infection, cryptococcal meningitis or invasive fungal diseases [35,36,37,38,39,40,41,42,43,44,45,46,47].

Autoimmune disorders are also frequent at diagnosis or during the course of the disease and may occur in one out of four patients. The clinical manifestations are variable: Behcet’s disease, immune cytopenia (e.g., acute autoimmune haemolytic anaemia), vasculitis, rheumatoid arthritis, peripheral neuropathy. Some of them may be life-threatening [48,49,50,51,52,53,54,55,56,57,58,59,60,61,62,63].

Extramedullary clinical presentations have been described and may reveal HCL: skin, bone, pleuro-pericardial or central nervous system (CNS) infiltration [48,64,65,66,67,68,69,70,71,72].

We propose the following “diagnostic box” in order to confirm the diagnosis and to rule out differential diagnoses, especially HCL-like disorders (Table 1).

The diagnosis is based on four criteria: (1) complete blood count (CBC) with morphologic assessment, (2) trephine bone marrow biopsy with dry marrow aspirates, (3) multiparametric flow cytometry (FCM) and, (4) identification of the BRAF^V600E^ mutation.

## 3. Complete Blood Count (CBC) with Careful Search for Hairy Cells (HCs)


-A careful examination of the PB smear is necessary for the identification of HCs. HCL patients usually have a low white blood count and the tumor burden is usually low.-CBC may show one or more cytopenias of varying severity: anemia, neutropenia or thrombocytopenia.-Lymphocytosis is uncommon [73].-Unlike HCL-like disorders, monocytopenia is often present in HCL: it can be masked by automated hematology analyzers, which frequently identify HCs as monocytes [74].-The morphology of HCs is characteristic. HCs have long, fine and circumferential villi, a mature and homogeneous chromatin, and occasional nucleoli. HCs are medium-sized lymphocytes with an abundant, poorly defined, weakly and heterogeneously basophilic cytoplasm. The cytoplasmic projections that give hairy appearance are narrow and circumferential. The nucleus is often round, oval, or kidney-shaped and the chromatin is dispersed with rare or inconspicuous nucleoli. Even if HCs are in low number in the PB, their identification has a strong value for the diagnosis.


## 4. Trephine Bone Marrow Biopsy with Dry Marrow Aspirates


-BM can be difficult to aspirate due to reticulin fibrosis.-As in the international consensus guidelines [75], we strongly recommend BM trephine biopsy (with immunostaining for CD20, DBA44, TRAP, VE-1, Cyclin D1 and Annexin-A1). It evaluates the degree of BM infiltration and is useful when assessing the response after treatment.-Medullary tumor infiltrate is commonly interstitial with honeycomb appearance [76].


## 5. Flow Cytometry (FCM)


-An immunophenotype can be carried out on PB or BM after mononuclear cell concentration by gradient density to improve sensitivity. HC can be localized very close to the monocyte gate. At the very least, a marker panel should combine markers for analysis of the B-cell lineage (CD19, CD20), for immunoglobulin light chain isotype restriction, and markers indicative of HCs (CD11c, CD25, CD103, CD123).-The four markers CD11c, CD25, CD103 and CD123 define the HCL immunophenotypic score [2] (one point is given to each expressed marker), which distinguishes HCL that are ≥3 in 98% of cases from other HCL-like disorders with a low score <3. CD103 expression could be negative, weak or with bimodal expression [77,78].-Differential diagnosis markers such as CD200, CD180, CD27 and CD26 could be useful in complex cases [79,80]. CD200 is often expressed in HCL but not in HCL-like disorders [81]. The added value to the HCL score of CD26 expression has been recently demonstrated [80].-The expression of CD38, present in one-third of HCL, confers a poor prognosis [82] as in CLL.-On the contrary, HCs are usually negative for CD5, CD10, CD23 and CD27.-FCM may show an unusual phenotype in some cases of HCL [83,84,85]. These abnormalities should not cast doubt on the diagnosis.


## 6. Cytogenetics and Molecular Analysis


-We do not recommend performing a karyotype at diagnosis of HCL, even if it may show several abnormalities. In fact, the presence or absence of these abnormalities will not modify the type of treatment.-We recommend BRAF testing for diagnosis. BRAF testing should be carried out whenever possible in the BM aspirate. The BRAF*^V600E^* mutation is present in more than 90% of HCL [3] and is usually absent in HCL-like disorders. The integration of molecular data is essential [80]. The method used to look for BRAF*^V600E^* mutation is left to the centers. In case of BRAF*^V600E^* negativity, an extended sequencing of BRAF (exon 11 and 15) is recommended because of the possible use of targeted treatments. Some patients have alternative BRAF mutations close to the valine in position 600, which impairs the specifically targeted BRAF*^V600E^* molecular testing [86].-Preservation of cells and serum in a cell bank is also recommended before any treatment for further analysis and a better understanding of the pathophysiology of the disease.


## 7. Other tests: Biochemistry, Viral Studies, Imagery

These tests are important to evaluate comorbidities, contra-indications of some treatments, and the impact of HCL on organ functions.


-LDH and β2 microglobulin have a prognostic value.-Screening for hemolysis (direct antiglobulin test, haptoglobin, unconjugated bilirubin and lactic dehydrogenase) and immunodeficiency risk by HIV, hepatitis B and C serology is also necessary.-Chest radiograph or computed tomography (CT) scan (chest abdomen and pelvis) is likewise necessary before any treatment.-Magnetic resonance imaging (MRI) can be necessary in case of symptomatic and extramedullary manifestations; it must be completed by histologic analysis of the involved tissues.-Although it may reveal some hypermetabolic lesions, especially in cases of extramedullary involvement [67,68,87,88], positron emission tomography (PET)/CT is not recommended at diagnosis by the experts.




**Prognostic factors**



Currently, while no prognostic factor modifies the choice of first-line treatment some studies have identified a poorer response to CDA [89,90] in the following cases:-Leukocytosis > 10 × 10^9^/L,-Bulky spleen extending >10 cm below costal margin,-Unmutated (UM) *IGHV* profile (>98%),-Use of the *VH4-34* repertory,-Mutation of the *TP53* gene-In case of relapse, it is necessary to perform:-Next-generation sequencing (NGS) with a review of different molecular abnormalities (*BRAF, MAP2K1, TP53*),-*IGHV* mutational status and *IGHV* repertory,-*TP53* status to evaluate prognosis and adapt targeted therapies.



**Assessment of response and measurable residual disease (MRD)**




-Complete response (CR), partial response (PR), stable disease (SD), and progressive disease (PD) are defined in the international guidelines [75] and must be used in clinical practice. CR is defined by resolution of palpable splenomegaly, near normalization of PB count and eradication of HC from the BM. The definition of responses is described in Table 2.



-The BM trephine biopsy should be performed between 4 and 6 months after administering CDA and after obtaining a clinical and hematological response with pentostatin.-Immunohistochemical (IHC) measurable residual disease (MRD) analysis of BM biopsy using B-lineage or specific HCL antibodies (VE-1, DBA.44, TRAP) is correlated to relapse [91,92,93]. The sensitivity of IHC (nearly 1% or less if dual-staining) is overcome by high-throughput technologies such as FCM and molecular analysis for detecting MRD [94,95]. Patient-specific RQ-PCR based on clone-specific IGH is very sensitive to quantify MRD but laborious [96]. Because BRAF*^V600E^* seems to be the primary event in HCL [97,98], an interesting alternative is allele-specific qPCR of DropletDigitalPCR (ddPCR) against BRAF*^V600E^* [19,95,99,100]. A recent consensus guideline insisted that MRD must be performed on the first pull BM aspiration because of poor medullary infiltration and limited PB involvement [101].


In elderly patients or in patients with severe comorbidities, BM evaluation is not mandatory. In these cases, a physical examination and CBC are sufficient to evaluate response after treatment. Achieving a hematologic complete response (HCR) is a good end point in this unfit population and quality of life should be evaluated and prioritized. Preliminary data on MRD after treatment combining rituximab (R) and vemurafenib suggest that undetectable MRD (uMRD) by PCR against BRAF*^V600E^* correlates with higher relapse-free survival (RFS) [19] and more durable responses [23,101,102,103,104]. In case of PR or CR with detectable MRD, the clinical interest of adding a treatment in order to achieve CR with uMRD (second course of PNA, second course of R, or another therapeutic option) has not been demonstrated. Achievement of CR with uMRD should be balanced with the toxicity of treatments used to eradicate MRD [75].

## 8. HCL-like Disorders


-As differential diagnosis between all these villous proliferations may be difficult, we propose here a simplified algorithm based on the integration (Figure 1 and Table 3) of clinical, morphologic, histologic, immunophenotypic and molecular data that are required to ensure the best classification, the correct diagnosis and therefore the best therapeutic management.



-The peripheral blood picture in HCL-V is monomorphic, with morphology of the abnormal lymphoid cells between hairy cells and prolymphocytes. The proportion of atypical lymphoid cells ranges from 20% to 95% and in most cases accounts for more than 50% of mononuclear cells. The cells are medium to large in size and have an abundant basophilic cytoplasm with circumferential hair-like cytoplasmic projections. The nucleus has a prominent vesicular nucleolus and a condensed chromatin. In HCL-V, the expression of CD25 is negative in all cases at diagnosis and the expression of CD103 and CD123 is usually dim or negative. The expression of CD11c is positive and bright.-Distinction between SDRPL and HCL-V may be challenging. However, cytomorphologic blood examination of SDRPL should find numerous villous lymphocytes (always > 20%, often > 60%). The nucleolus is present but not prominent [105]. Immunophenotypic features find high expression of CD11c and CD180, CD27 negativity and diminished expression of CD200 [79,106]. The CD200/CD180 ratio is equal or below 0.5 [79]. Unlike HCL, HCL-V and SDRPL, the splenic involvement of SMZL affects the white pulp and not the red pulp of the spleen.-To avoid diagnostic splenectomy, SMZL may be distinguished from SDRPL or HCL-V thanks to cytomorphologic examination: SMZL leukemic cells have more clumped chromatin, less prolonged villi with a broad base, and villous lymphocytes represent less than 20% of lymphomatous circulated cells [107]. The immunophenotypic profile differs due to diminished expression of CD11c, positivity of CD27, and medium expression of CD180 and CD200 [79,106]. Mutational landscape is dominated by *KLF2* and *NOTCH2* mutations (in about 40% and 20% of cases, respectively) [108,109,110,111]. The *IGHV* repertory is biased with overrepresentation of VH1-2 in one-third of cases [112]. The nucleolated form of SMZL that could correspond to the evolution of the indolent form is now classified with HCL-V and B-PLL in the SBLPN group [113,114].-The Japanese variant form of hairy-cell leukemia (HCL-JV) is more frequent in Japan than classical HCL. However, this entity is very rare; only 17 cases have been published so far, and they are closer to SDRPL [115]. The median age at diagnosis is 75 years. Men are more affected (sex ratio 3.2:1). Splenomegaly is recurrent, lymph nodes are rare and the HCs do not have prominent nucleoli. As in HCL-V and SDRPL, lymphocytosis is frequent, and there is no expression of CD25 [115]. Further investigations should be carried out to confirm the specific pattern of HCL-JV or if they are related to SDRPL.


## 9. Treatment

We encourage inclusion in clinical trials whenever possible both in front-line and relapsed/refractory (R/R) settings. Inclusion in real-life registries and retrospective studies is also recommended whenever possible, because HCL is rare, and only collaborative studies can increase our knowledge and improve clinical outcomes for patients. As in all indolent B-cell lymphoproliferative disorders, HCL and HCL-like disorders should be treated only if one or more of the following criteria are present: (1) Hb < 11 g/dL and/or platelets < 100 G/L and/or neutrophils < 1 G/L, (2) symptomatic organomegaly, (3) constitutional symptoms (fever, weight loss, night sweats), (4) recurrent infections. If none of these criteria are present, we recommend a ‘watch and wait’ strategy. If a treatment is required, our expert panel recommends the following therapeutic algorithms for HCL (Figure 2), HCL-V (Figure 3) and SDRPL (Figure 4). Details on the therapeutic regimens are given in Table 4.



**Hairy-Cell Leukemia**




**First-line treatment in symptomatic HCL patients**



-**In first-line and fit patients, we recommend CDAR regimen** with CDA plus rituximab (R). The data with P plus rituximab are limited [115]. CDA plus R was demonstrated to be more effective than PNA in monotherapy, allowing to reach undetectable MRD (uMRD) in most patients and to achieve more durable responses.-The simultaneous combination of R + CDA (i.e., R started at day 1 of CDA) was shown to be superior to the sequential schedule of CDA followed by R (i.e., R started 6 months later). R is given at day 1 of CDA for 8 weekly infusions [116]. The overall response rate (ORR) and the CR rate were both 100%, with 97% of patients reaching uMRD.-The subcutaneous (SC) formulation of CDA (0.1–0.14 mg/kg/d once per day for 5 days) is favored as compared to the intravenous (iv) formulation (0.1 mg/kg/d as continuous intravenous infusion for 7 days) [117,118].



**Cladribine (CDA) or pentosatin (P) in monotherapy**



-PNAs in monotherapy remain an option as the patients may achieve a durable response even if they do not reach uMRD. CDA is given subcutaneously at 0.1–0.14 mg/kg/d once per day for 5 days. Pentostatin is given intravenously at 4 mg/m^2^ once every 2 weeks for one year. If there is no response at 6 months, P should be stopped and another treatment should be discussed [11,119].-In the recently updated real-life and retrospective French HCL cohort, the ORR and the CR rate were 100% and 83%, respectively, in patients receiving PNA in the front-line setting. Median RFS was 163 months with CDA and 159 months with P [12]. The five-year overall survival (OS) was 97% for patients treated with CDA and 86% for those treated with P. Similarly, in an Italian study including 513 patients treated with CDA in first line, the ORR and CR rate were 91.8% and 65.3%, respectively [120]. Median RFS was 12.2 years and five-year OS 95.3%.



**Particular situations**


For unfit patients not able to receive CDA + R, PNA in monotherapy is a good option. Two or three days (instead of five) of subcutaneous CDA may be used in very elderly people. If patients are unfit to receive PNA alone, BRAFi +/− R or IFNα are alternative choices [9,17]. R alone is not recommended because responses are partial and short-lasting [121,122,123,124,125,126,127].

In case of active infection, we recommend to delay the start of HCL treatment until infection is controlled. If delaying the start of treatment is not possible, the use of a bridging therapy with low-dose BRAFi is our recommendation due to the absence of myelotoxic and myelosuppressive effects.

It has been demonstrated that BRAFi can rapidly improve the neutrophil count and thereby facilitate infectious disease control [128]. Once infection is controlled, a switch to CDA + R is recommended. In the same way, in case of pandemic periods such as COVID-19, it is preferable to delay HCL treatment and, if it is not possible, to recommend BRAFi as a bridging therapy [129]. However, a recently published study on the largest cohort of HCL/HCL-V patients with COVID-19 infection showed a low mortality rate (about 1%) [130].

In the rare case of HCL occurring during pregnancy, we recommend delaying treatment until after childbirth. If delaying HCL therapy is not possible, use of IFNα is preferable. Of note, there are no data on BRAFi during pregnancy, even if some case reports have been published in patients with melanoma [131].


-In case of renal insufficiency, CDA is contra-indicated if clearance is ≤ 50 mL/min and P if clearance is < 60 mL/min because of a lack of data and a potential risk of increased toxicity as PNAs and their metabolites are mainly excreted renally. There are no specific data in the literature for patients with HCL and chronic kidney disease. In this case, in order to achieve CR with uMRD, particular caution is advised and risks/benefits should be carefully evaluated if administration of PNA is considered; alternatively, if BRAF is mutated, vemurafenib could be used.



**Second line treatment**



-In case of relapse, a new complete work-up is needed, including a complete physical examination, CBC, trephine BM biopsy, and FCM. We also recommend BRAF testing again, the study of *IGHV* mutational profile, use or not of the *VH4-34* repertory and the *TP53* status. UM *IGHV*, *VH4-34* usage and *TP53* alterations confer an unfavorable prognosis in HCL [89,90,132]. IGHVs are unmutated in 10% of HCL cases, which are associated with resistance to CDA, shorter event-free survival (EFS), high white blood cell count, splenomegaly, and p53 dysfunction [89]. *VH4-34* usage occurs in 10% of HCL and 40% of HCL-V and is associated with high white blood cell count, UM IGHV, a lower response rate to CDA, and shorter PFS and OS [90]. *TP53* mutations are present in 0–2% of HCL and 30% of HCL-V and confer a poor prognosis in HCL [132,133,134].-Type of first-line treatment, duration of response, quality of response, toxicities of previous treatment, age and comorbidities must be considered for the choice of second-line treatment in HCL.


**In case of response lasting more than five years and if the patient is fit, a treatment combining a CDAR is recommended again.** We recognize that there are no data in the literature supporting re-treatment with CDA plus R rather than a switch to targeted therapies such as vemurafenib plus R. Because of the progressive chemo-refractoriness after several lines of PNAs with lower and shorter responses at each line, we recommend the addition of R instead of CDA alone [12,13,14].

In case of response between two and five years, the different options should be considered based on the number and severity of associated comorbidities and clinician judgment. The options to be discussed are CDA plus R or vemurafenib plus R.


**In case of primary refractory disease or in case of short response lasting less than two years after first-line therapy, we recommend the use of vemurafenib plus R.**


This off-label treatment has been demonstrated to be effective in the R/R setting and patients treated with this combination have achieved deep responses with uMRD [19]. In a single-center phase 2 study including 30 R/R HCL patients, vemurafenib was given at 960 mg twice daily for 8 weeks and R at 375 mg/m^2^ for eight infusions over 18 weeks (started at day 1 of vemurafenib). The ORR and CR rate were both 87%, with 65% of CR patients achieving uMRD (MRD with PCR BRAF*^V600E^* in PB and BM). PFS was 78% after median follow-up of 37 months. Practitioners must be aware of cutaneous (rash, photosensitivity, basal-cell carcinoma, squamous-cell carcinoma, melanoma), articular (arthralgia, arthritis), ocular, pancreatic and hepatic (elevation of lipasemia and transaminitis) adverse events. Electrocardiogram (ECG) must be regularly performed because of possible prolongation of the QT interval. Dose reductions of vemurafenib are frequent, as it may be effective at lower doses [128,135]. Vemurafenib is not myelotoxic and less immunosuppressive than PNA. The combination of R + vemurafenib should be preferred to BRAFi or MEKi alone because of their lower and shorter responses in monotherapy, with no uMRD [17,18]. In case of relapse after R + vemurafenib, re-treatment with the same combination is possible and may be effective.

**Moxetumomab pasudotox (Moxe),** an anti-CD22 immunotoxin containing a toxin of *Pseudomonas aeruginosa*, is no longer available in most countries [136]. In countries where it is available, Moxe is a good alternative option as it has demonstrated deep and long-lasting responses in R/R HCL [23,137,138].

In the multicenter pivotal trial including 80 R/R HCL patients, Moxe was administered at 40 µg/kg on days 1, 3 and 5 for a maximum of six cycles of 28 days. The ORR was 75%, CR was 41%, and the durable CR rate, which was the primary endpoint of this study and corresponds to CR lasting more than 180 days, was 36%. Eighty-two percent of CR patients were MRDu (MRD assessed with IHC on BM). Median PFS was 71.7 months. Patients receiving Moxe must be monitored for capillary leak syndrome and hemolytic and uremic syndrome. As vemurafenib, Moxe is not myelotoxic.

Another interesting option in this setting is the experimental combination of a BRAFi with a MEKi. We recommend using dabrafenib + trametinib as described by *Kreitman* et al. [20]. In a phase 2 study, 55 R/R HCL patients received continuous dabrafenib 150 mg twice daily with trametinib 2 mg once daily until disease progression or unacceptable toxicity. The ORR and CR rates were 89.0% and 65.5%, respectively. One quarter of CR patients achieved MRDu (measured with FCM in PB and BM). At 24 months, PFS and OS were 94.4% and 94.5%, respectively. However, this combination produced adverse events (63.6% grade 3–4 in the phase 2 study: pyrexia, pneumonia, neutropenia, and hyperglycemia).

## 10. Third Line Treatment and beyond


-In patients in third line and beyond, we do not recommend the use of PNA again, especially if recurrence of HCL has occurred within 2–5 years. The use of several courses of PNA is associated with cumulative toxicities linked to immunosuppression, especially infectious diseases and second primary malignancies [12,16,139,140,141,142,143,144,145,146]. HCL becomes less responsive to PNA at each line, with lower response rates and shorter PFS [12,13,14].


**If patients have not previously received BRAFi plus rituximab or MEKi,** we recommend the use of vemurafenib and rituximab if the patient is BRAF^V600E^.

**If patients have previously received BRAFi,** we recommend ibrutinib, or venetoclax +/− R for BRAF*^V600E^* HCL, and ibrutinib or venetoclax +/− R (Figure 2). The latter combination has been shown to be effective in first line and in R/R cases of HCL [147,148], but patients should be carefully monitored for toxicity due to the expected prolonged immunosuppression. Moreover, historical treatments such as IFNα-2a [9] or splenectomy [8] should also be kept in mind in the absence of contra-indication.

**Ibrutinib**, a Bruton’s tyrosine kinase (BTK) inhibitor (BTKi), is already used in CLL, mantle cell lymphoma, Waldenström macroglobulinemia, primary central nervous system lymphoma and splenic marginal zone lymphoma. As with ibrutinib, BTKi has joined the therapeutic armamentarium in HCL and HCL-like disorders [5]. Experience with other BTKis such as acalabrutinib, zanubrutinib or pirtobrutinib is anecdotal. In a phase 2 study, 37 patients (28 R/R HCL, two previously untreated HCL-V, and seven R/R HCL-V) with a median number of treatment lines of 4 (0–12) were treated with ibrutinib at 420 mg/d (n = 24) or 840 mg/d (n = 13), continuously until disease progression or unacceptable toxicity [21]. The ORR was 24% at 32 weeks, 36% at 48 weeks, and 54% at any time. CR was rare. Estimated 36-month PFS and OS were 73% and 85%, respectively. Interestingly, there were no significantly different outcomes between HCL and HCL-V patients, even if the study was not designed to compare these two populations and even if the number of patients is small. Moreover, there was no statistically significant difference between the two ibrutinib doses. The toxicity profile was classic for BTKi: atrial fibrillation, atrial flutter, arterial hypertension, and bruising. Even if a cross-comparison between the different clinical trials is not pertinent, ibrutinib seems to be less effective than the three previously described options: R + vemurafenib, Moxe, dabrafenib + trametinib. The expert panel recommends ibrutinib only for patients who relapsed after/had no access to/were intolerant to all these options. A recently published study showed a similar efficacy and toxicity profile for zanubrutinib, another BTKi, but data are still preliminary [22].

B-cell lymphoma-2 (Bcl-2) is highly expressed in HCL [149]. Venetoclax, a bcl-2 inhibitor (Bcl-2i), restores apoptosis in cancer cells and is approved in the treatment of CLL and AML [150,151]. Recently, the cytotoxicity of venetoclax on HCL cells was demonstrated in vitro [152]. Moreover, in a recently published study by Tiacci et al*.,* venetoclax was used in six HCL patients who had previously received R + vemurafenib as their last therapy [153]. Venetoclax was given for up to twelve 28-day cycles. Two patients achieved CR with detectable MRD, one patient PR, one patient a minor response, one patient no response and the last one a progressive disease. R was added to 3/6 patients for 3 to 8 cycles, improving the response for three patients.

## 11. Supportive Care in All Cases

For patients who have received a treatment for HCL or HCL-like disorder, we recommend a prophylaxis against *Pneumocystis jirovecii* with cotrimoxazole, and a prophylaxis against zona and herpes zoster viruses with valaciclovir or aciclovir. When PNAs are used, we recommend starting these molecules two weeks after initiation of PNAs, because of possible cutaneous adverse events. Prophylaxis must be continued for at least six months and until the CD4+ T-cell count is >0.2 G/L. In case of transfusion for a patient who has received PNAs, irradiation of the blood products could be discussed. We recommend the use of G-CSF in case of febrile neutropenia, in combination with large-spectrum antibiotics. Vaccines against influenza, SARS-CoV-2, and *Streptococcus pneumoniae* should ideally be realized before starting treatment if possible, especially in elderly patients with comorbidities. Prevention of second primary malignancies, especially cutaneous cancers, is recommended because of the high risk of second cancers in HCL patients, regardless of the treatment received [12,15,17,139,140,146,154,155,156,157]. We recommend a regular (at least once a year) complete physical examination with a dermatologist.


**HCL-like disorders**



**HCL-V**


The treatment criteria are identical to those for HCL (Figure 3). If none of the criteria are present, a ‘watch and wait’ strategy is recommended. Patients are often refractory to PNA monotherapy, which is why we recommend the combination of R + CDA in first line. As in classic HCL, we propose the use of the same simultaneous R + CDA schedule published by Chihara et al.: cladribine with R started at day 1 of CDA, with eight-weekly doses of R [158]. Among 20 patients, including eight previously untreated, the CR rate was 90% and the uMRD rate 80% (FCM in PB and BM and IHC). Five-year PFS was 63.3% and five-year OS was 74%. In case of lymphocytosis, in order to diminish the occurrence of infusion reaction to R, it should be started once the lymphocyte count returns to a normal range after CDA. In R/R HCL-V, a MEKi such as trametinib may be used if the *MAP2K1* mutation is present [159]. Another option for R/R HCL-V is a BTKi such as ibrutinib, as efficacy seems to be similar between HCL and HCL-V [21]. In countries where previously described targeted therapies are not available, the combination of R + bendamustine may be considered, with some good results in HCL and HCL-V [160,161]. Regarding Moxe, in the pivotal trial, only three HCL-V patients were included and none achieved CR [137]. Therefore, with current data, we cannot recommend Moxe for HCL-V patients.

## 12. SDRPL

The treatment criteria are those for HCL (Figure 4). If none of the criteria are present, a ‘watch and wait’ strategy is recommended. Recommendations and therapeutic algorithms are difficult to establish in SDRPL because of the paucity of published studies due to the rarity of disease. In first line, if the patient is fit and has no comorbidity, we recommend splenectomy. This strategy often leads to prolonged responses [162,163]. If the patient is not eligible for splenectomy, if he refuses it, or in R/R cases, immunotherapy with R, chemotherapy or immunochemotherapy may be used. Our expert panel proposes R + CDA, R + bendamustine, or R alone. In a study including 17 SDRPL patients, 11 were treated with splenectomy and six received chemotherapy followed by splenectomy. Five-year OS was 93% [162].

## 13. Conclusions

The immunochemotherapeutic approach, with the combination of CDA + R, is now the gold standard for the treatment of HCL in the front-line setting, allowing achievement of CR with MRDu for most patients and prolonged PFS. Progress in the pathophysiology of HCL has opened the way for more widespread use of targeted therapies in HCL and HCL-like disorders. PNA should not be used for more than two lines because of the cumulative toxicities and the progressive refractoriness of HCL to this therapeutic class. Therefore, the combination of anti-CD20 mAb with BRAFis such as R + vemurafenib is recommended by our expert panel for R/R HCL harboring the BRAF*^V600E^* mutation. Moxe if available, the combination of BRAFi + MEKi, BTKi, or Bcl-2i are other options for R/R HCL. Inclusion in clinical trials and in national/international registries is strongly encouraged. We think that new treatments such as anti-CD22 chimeric antigen receptor T-cells will change this therapeutic algorithm in the following decade.

## Figures and Tables

**Figure 1 cancers-16-02185-f001:**
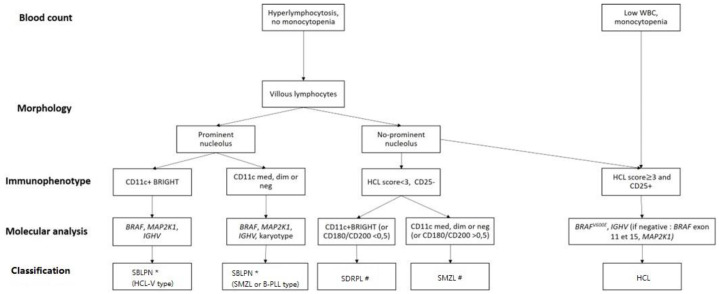
Diagnostic algorithm for hairy-cell leukemia (HCL) and HCL-like disorders: HCL variant (HCL-V), splenic diffuse red pulp lymphoma (SDRPL), splenic marginal zone lymphoma (SMZL), B-cell prolymphocytic leukemia (B-PLL). # Determination of % of villous lymphocytes/total abnormal lymphocytes (if≥20% → SDRPL). * Exclusion of MCL or CLL related-PLL if CD5+: karyotype t(11;14) or SOX11/CCND1 and Matutes RMH score. Dim: diminished expression, med: medium expression; neg: negative expression.

**Figure 2 cancers-16-02185-f002:**
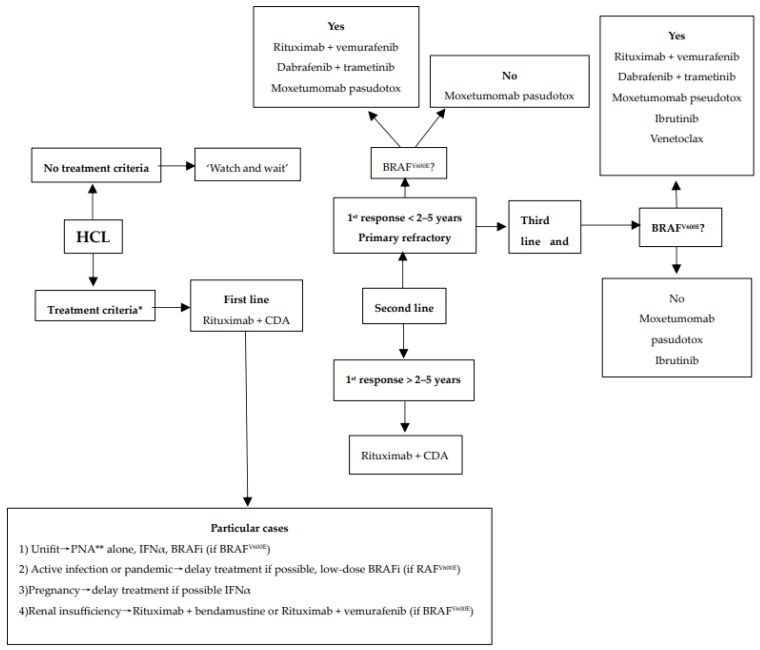
Therapeutic algorithm for treatment of patients with hairy-cell leukemia (HCL). * one or more of the following criteria: (1) Hb < 11 g/dL and/or platelets < 100 G/L and/or neutrophils < 1 G/L, (2) symptomatic organomegaly, (3) constitutional symptoms (fever, weight loss, night sweats), (4) recurrent infections. ** sc cladribine or iv pentostatin.

**Figure 3 cancers-16-02185-f003:**
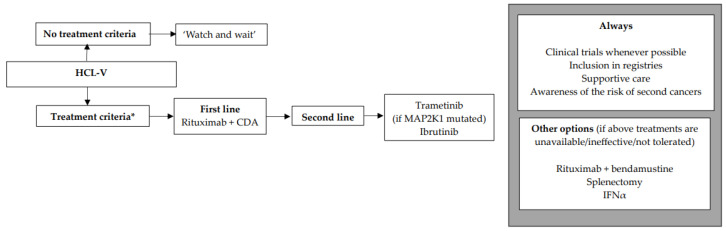
Therapeutic algorithm for treatment of patients with hairy-cell leukemia variant (HCL-V). * one or more of the following criteria: (1) Hb < 11 g/dL and/or platelets < 100 G/L and/or neutrophils < 1 G/L, (2) symptomatic organomegaly, (3) constitutional symptoms (fever, weight loss, night sweats), (4) recurrent infections.

**Figure 4 cancers-16-02185-f004:**
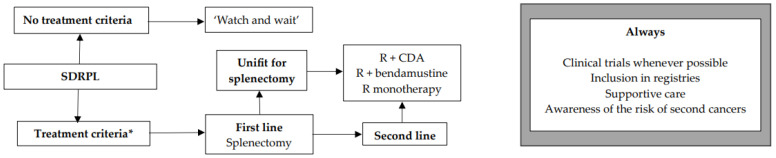
Therapeutic algorithm for treatment of patients with splenic diffuse red pulp lymphoma (SDRPL). * one or more of the following criteria: (1) Hb < 11 g/dL and/or platelets < 100 G/L and/or neutrophils < 1 G/L, (2) symptomatic organomegaly, (3) constitutional symptoms (fever, weight loss, night sweats), (4) recurrent infections.

**Table 1 cancers-16-02185-t001:** Baseline and pretreatment evaluation of HCL patients. # targeted molecular analysis: qPCR or ddPCR. CBC: complete blood count, IGHV: immunoglobulin heavy chain gene, IHC: immunohistochemistry, MRI: magnetic resonance imaging, PET CT: positron emission tomography computed tomography, MRD: measurable residual disease.

Diagnosis	General Practice
**For diagnosis/classification**	
CBC and blood smear revision (nucleolus)	All cases
Immunophenotyping (HCL score assessment)	All cases
BRAF^V600E^ mutational status	All cases (IHC or molecular analysis #)
Bone marrow trephine biopsy	All cases
IGHV repertory and mutational status	Recommended if BRAF^WT^ or if R/R cases
Extended mutational landscape (BRAF exon 11 and 15, MAP2K1, KRAS, NRAS, HRAS)	Recommended if BRAF^WT^
TP53 mutational status	Recommended if R/R cases
Cytogenetic analysis	Not recommended
**Pretreatment assessment**	
History, physical examination (performans status)	All cases
Creatinine clearance	All cases
Direct antiglobulin test, haptoglobin, unconjugated bilirubin and lactic dehydrogenase	All cases
Transaminases, hepatitis B and C serology	All cases
Preservation of cells and serum in a tumor bank	Recommended
Chest radiograph or CT scan (chest abdomen and pelvis)	Recommended
MRI/PET CT scan	In symptomatic patients with unusual extramedullary localizations
**Assessment for MRD**	Recommended

**Table 2 cancers-16-02185-t002:** Response criteria in HCL.

Response	Definition
Complete response (CR)	Regression of splenomegaly on physical examination. Near normalization of blood count without transfusion: Hb > 11 g/dL, platelets > 100 G/L, neutrophils > 1.5 G/L. Absence of hairy-cell on blood smear and bone marrow examination.
Complete response with MRD negativity (CR MRD-)	CR and MRD negativity by IHC or FCM on bone marrow examination.
Partial response (PR)	Regression of at least 50% of splenomegaly on physical examination. Near normalization of blood count without transfusion: Hb > 11 g/dL, platelets > 100 G/L, neutrophils > 1.5 G/L. Regression of at least 50% of bone marrow infiltration by hairy cells.
Stable disease (SD)	Patients who do not met criteria for CR, PR ou PD.
Progressive disease (PD)	Increase of at least 25% of splenomegaly on physical examination and/or decline of at least 25% of hematological parameters and/or increase of symptoms.
Hematologic complete response (HCR)	Regression of splenomegaly on physical examination. Near normalization of blood count without transfusion: Hb > 11 g/dL, platelets > 100 G/L, neutrophils > 1.5 G/L.
Hematologic partial response (HPR)	Regression of at least 50% of splenomegaly on physical examination. Near normalization of blood count without transfusion: Hb > 11 g/dL, platelets > 100 G/L, neutrophils > 1.5 G/L.

**Table 3 cancers-16-02185-t003:** Clinical and biological characteristics of HCL and HCL-like disorders. The cut-off values for positivity of markers are variable between studies. ND: no data, +/− means that it can be positive or negative.

		HCL	HCLv	SDRPL	SZML
**Epidemiology**	Incidence	0.3	0.2	ND	0.2
	Sex ratio M/F	4	1.6	1.6	0.5
	Median age at diagnosis	55	70	77	62
**Clinical data**	Infections	Yes	Yes	Yes	Yes
	Splenomegaly	Yes	Yes	Yes	Yes
	B symptoms	rare	rare	1/3	1/4
**Complete blood count**	Cytopenia	Yes	Yes	Yes	Yes
	Monocytopenia	Yes	No	No	No
	Lymphocytosis	≤10%	≥90%	≥50%	≥50%
**Cell morphology**	Villi	Long, fine and circumferential	Long, fine and circumferential, sometimes shaggy	Long, large, broad base and polar	Small and polar
	Nucleoli	Occasional, inconspicuous	Constant, prominent	Occasional	Small
	Chromatin	Mature, homogene	Mature, homogene	Condensed	Condensed
**Immunophenotype**	CD5	-	-	-	+/−
	CD11c	+Bright	+(63%)	+(>90%)	+dim (33–67%)
	CD23	-	-	-	-
	CD25	+Bright	−(>90%)	−(>90%)	−(78–88%)
	CD27	−(>90%)	ND	−(81%)	+(89%)
	CD103	+Bright	+(65–100%)	−(62–84%)	−(>90%)
	CD123	+Bright	−(60%)/dim	−(50–84%)	−(75%)/dim
	CD180	+	ND	+ Bright	+
	CD200	+Bright	-	-	-
**Histology**	Bone marrow infiltration	fibrosis, intrasinusoidal	interstitial, intrasinusoidal	interstitial, intrasinusoidal	intrasinusoidal, nodular
	Spleen infiltration	Red pulp	Red pulp	Red pulp	White pulp
**Immunohistochemistry**	Annexin A1	+	-	-	-
	DBA 44	+	+	+	+
	VE 1	+	-	-	-
	Cyclin D1	+	+/−	-	-
**Cytogenetics**	Abnormal karyotype	40%	rare	30%	80%
	Abnormalities	del(17p), del(7q), +12	del(17p), del(7q), +12, complex	del(7q), +3, +18, complex	del(7q), +3, +18
**IGHV**	Mutated	83–90%	46–73%	79%	59–68%
	Repertory	VH3-30, VH3-23, VH4-34	VH4-34	VH4-34, VH3-23	VH1-2, VH4-34
**Genetics**	ARID1A	4–5%	4%	8%	ND
	BCOR	0–5%	0%	24%	2%
	BRAF V600E	70–100%	0%	0–2%	0–2%
	CCND3	0%	13%	21–24%	13%
	CDKN1B	10–16%	0%	4%	ND
	CREBBP	5–6%	12–25%	ND	ND
	KDM6A	0–2%	12–50%	2%	0%
	KLF2	13–16%	0%	2–4%	20–30%
	KMT2C	15%	25%	ND	ND
	MAP2K1	0–22%	38–42%	7–12%	0%
	MYD88	0%	ND	0%	9%
	NOTCH1	4–13%	0%	2%	9%
	NOTCH2	0–4%	0%	10%	17–25%
	TNFAIP3	0%	ND	0%	20%
	TP53	2–28%	8–38%	5–15%	13–25%
	U2AF1	0%	13%	ND	ND

**Table 4 cancers-16-02185-t004:** Therapeutic regimens for HCL and HCL-like disorders.

Regimen	Details
**CDA**	sc 0.1–0.14 mg/kg/d once per day for 5 days
**CDA + R**	CDA sc 0.1–0.14 mg/kg/d once per day for 5 days + R iv 375 mg/m^2^, 8 weekly infusions, the first one started at day 1 of CDA
**Pentostatin (P)**	iv 4 mg/m^2^ once every 2 weeks for one year. If there is no response at 6 months, P should be stopped and another treatment should be discussed
**Vemurafenib**	960 mg twice daily for 16–18 weeks
**Vemurafenib + R**	vemurafenib 960 mg twice daily for 8 weeks + R 375 mg/m^2^ 8 infusions over 18 weeks (started at day 1 of vemurafenib)
**Dabrafenib + Trametinib**	dabrafenib 150 mg twice daily + trametinib 2 mg once daily until disease progression or unacceptable toxicity
**Trametinib**	2 mg once daily until disease progression or unacceptable toxicity
**Moxetumomab pasudotox**	iv 40 µg/kg on days 1, 3 and 5 for a maximum of six cycles of 28 days
**Ibrutinib**	420 mg/d or 840 mg/d until disease progression or unacceptable toxicity
**Venetoclax**	400 mg/d (after a ramp-up phase), maximum 12 28-day cycles
**Bendamustine + R**	bendamustine iv 90 mg/m^2^ at day 1 and day 2 + R iv 375 mg/m^2^ at day 1; 6 28-day cycles

## Data Availability

Not applicable.

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
