# Peer review of "Recommendations for the Management of Patients with Hairy-Cell Leukemia and Hairy-Cell Leukemia-like Disorders: A Work by French-Speaking Experts and French Innovative Leukemia Organization (FILO) Group"

_cancers, 2024, doi:10.3390/cancers16122185_

Round 1

Reviewer 1 Report

Comments and Suggestions for Authors

The authors summarize recommendations for the diagnostic and therapeutic management of patients with hairy cell leukemia and similar disorders. A table with the diagnostic characteristics and four figures with diagnostic and therapeutic algorithms are very useful.

Specific Points of Criticism and Suggestions for Alterations:

(1)  While the English is very good (translated by a professional translator), a few minor errors should be corrected, for example line 40:  “résultats” (= French), line 298: “different responses”, line 368: “a switch to”, line 530: “foro”

(2)  Title:  The title is curious. Is this the translation of recommendations originally written in French? Or are these recommendations in English specifically for french-speaking clinicians (inside and outside of France)? This is explained clearly in the manuscript. So either the circumstances should be explained or the title changed. Lines 88-90: Why now in English?

(3)  Citations:  The mode of citations is unusual. For example on line 107 commonly it is written "[30-33]" (instead of "[30][31][32][33]").

Further good examples: 

Line 116: "[35-47]"

Line 121: "[48-62]"

Line 452: „ [12,16,141-148]“ - and elsewhere.

(4)  Line 113 and line 294:  "enlarged lymph nodes" (or) "lymphadenopathy" are more appropriate (instead of only "lymph nodes" = clinical slang).

(5)  Table 1:  This table is actually an unfocused figure. Better to put the content of this "Figure" in a “real” table in text form.

Also not elsewhere in the manuscript defined abbreviations in this table should be annotated in the legend to the table.

(6)  Line 159:  "An immunophenotype" (not "A phenotype").

(7)  Lines 178-179:  "We do not recommend ...". Why not? explain.

(8)  Tables and 3 are mixed up. The titles are correct, but the tables themselves (content) are wrong.

(9)  Table 2:  The real Table 2 should be cited in the text prior to page 6 and should also be placed earlier in the paper.

(10)  Line 233:  These antigens are not "specific" for HCL as they are also found on normal cells. I propose to use the terms "associated with" or "indicative of".

(11)  Table 2:  While this table with the clinical and biological chracteristics is overall excellent, the reader would benefit from arrangement of the antigens in the “Immunophenotype section” and of the genes in the “Genetics section” in ascending numerical order and in alphabetical order, respectively. For example, if I am interested specifically in the gene BCOR I have to look through all the whole list while in alpabetical order it would be much easier.

Explain in the legend to this table what in the immunophenotype and in the immunohistochemistry +, +/- and - means (cut-offs for the percentages).

In the cytogenetic nomenclature deletions are written as follows: "del(17p)" or "del(7q") - (and not "del17p" or "del7q").

Genes KDM6A and CREBBP:  in English the decimals are separated by a period and not by a comma (12.5% versus 12,5%). Just write 12%.

(12)  Line 428:  Maybe specific:  "... is no longer available in most countries ... due to"? (toxicity? costs? inefficiency?).

(13)  Any outlook (at the end of Discussion)? A glimpse into the future?

Author Response

Reviewer 1

The authors summarize recommendations for the diagnostic and therapeutic management of patients with hairy cell leukemia and similar disorders. A table with the diagnostic characteristics and four figures with diagnostic and therapeutic algorithms are very useful.

Specific Points of Criticism and Suggestions for Alterations:

(1)  While the English is very good (translated by a professional translator), a few minor errors should be corrected, for example line 40: “résultats” (= French), line 298: “different responses”, line 368: “a switch to”, line 530: “foro”

Many thanks. We corrected these errors.

(2)  Title:  The title is curious. Is this the translation of recommendations originally written in French? Or are these recommendations in English specifically for french-speaking clinicians (inside and outside of France)? This is explained clearly in the manuscript. So either the circumstances should be explained or the title changed. Lines 88-90: Why now in English?

We corrected and we agree with the comment.

We have changed the title of the manuscript which is now:

“Recommendations for the management of patients with hairy cell leukemia and hairy cell leukemia-like disorders: a work by French speaking experts and members of the French Innovative Leukemia Organization (FILO) group.

(3)  Citations:  The mode of citations is unusual. For example on line 107 commonly it is written "[30-33]" (instead of "[30][31][32][33]").

Further good examples: 

Line 116: "[35-47]"

Line 121: "[48-62]"

Line 452: „ 12,16,141-148]“ - and elsewhere.

We corrected in the text the mode of the citations in order to harmonize the manuscript. This is now, as an example [30,3,32,33].

(4)  Line 113 and line 294:  "enlarged lymph nodes" (or) "lymphadenopathy" are more appropriate (instead of only "lymph nodes" = clinical slang).

Thanks again, we corrected.

(5)  Table 1:  This table is actually an unfocused figure. Better to put the content of this "Figure" in a “real” table in text form.

Also not elsewhere in the manuscript defined abbreviations in this table should be annotated in the legend to the table.

We added and modified the Table 1 according to your comments. Table 1 is now clearer.

(6)  Line 159:  "An immunophenotype" (not "A phenotype").

We changed.

(7)  Lines 178-179:  "We do not recommend ...". Why not? explain.

We explained: In fact, the presence or absence of these abnormalities will not modify the type of treatment.

(8)  Tables and 3 are mixed up. The titles are correct, but the tables themselves (content) are wrong.

We corrected.

(9)  Table 2:  The real Table 2 should be cited in the text prior to page 6 and should also be placed earlier in the paper.

Sorry. We changed and adapted according to your comments.

(10)  Line 233:  These antigens are not "specific" for HCL as they are also found on normal cells. I propose to use the terms "associated with" or "indicative of".

We agree and we changed.

(11)  Table 2:  While this table with the clinical and biological chracteristics is overall excellent, the reader would benefit from arrangement of the antigens in the “Immunophenotype section” and of the genes in the “Genetics section” in ascending numerical order and in alphabetical order, respectively. For example, if I am interested specifically in the gene BCOR I have to look through all the whole list while in alpabetical order it would be much easier.

Explain in the legend to this table what in the immunophenotype and in the immunohistochemistry +, +/- and - means (cut-offs for the percentages).

In the cytogenetic nomenclature deletions are written as follows: "del(17p)" or "del(7q") - (and not "del17p" or "del7q").

Genes KDM6A and CREBBP:  in English the decimals are separated by a period and not by a comma (12.5% versus 12,5%). Just write 12%.

Many thanks again. We changed.

(12)  Line 428:  Maybe specific:  "... is no longer available in most countries ... due to"? (toxicity? costs? inefficiency?).

We have modified and clarified.

(13)  Any outlook (at the end of Discussion)? A glimpse into the future?

We added as suggested.

“We think that new treatments like anti-CD22 Chimeric Antigen Receptor T-cells will change this therapeutic algorithm in the following decade.”

Reviewer 2 Report

Comments and Suggestions for Authors

I truly enjoyed reading the recommendations by French speaking countries for the diagnosis and management of hairy cell leukemia. I have several minor comments aimed at further improving the manuscript:

1) in the title and later in the text it should be clear that the recommendations were made by some specific organization or body representative of French speaking countries. I suggest to clearly state this. Also, probably the title should state French speaking countries as a phrase instead of French speaking

2) were these recommendations previously published somewhere else in the earlier version. Where? Which version is the currently presented one? 

3) Section on recommended therapies would benefit from the table summarizing clearly target sub-population, dosing regimen with details such as dose, how many days, how many cycles, etc.

Author Response

Reviewer 2

I truly enjoyed reading the recommendations by French speaking countries for the diagnosis and management of hairy cell leukemia. I have several minor comments aimed at further improving the manuscript:

1) in the title and later in the text it should be clear that the recommendations were made by some specific organization or body representative of French speaking countries. I suggest to clearly state this. Also, probably the title should state French speaking countries as a phrase instead of French speaking

We changed the title, as suggested by the reviewer 1. Recommendations for the management of patients with hairy cell leukemia and hairy cell leukemia-like disorders: a work by French speaking experts and French Innovative Leukemia Organization (FILO) group.

2) were these recommendations previously published somewhere else in the earlier version. Where? Which version is the currently presented one? 

These recommendations were not previoulsy published, but were orally presented at our last meeting of the Société Française d’Hématologie (March 2024, Paris). We added this information.

3) Section on recommended therapies would benefit from the table summarizing clearly target sub-population, dosing regimen with details such as dose, how many days, how many cycles, etc.

We agree and we clarified (Table 4).